# Bridgeless Boost Converter with an Interleaving Manner for PFC Applications

**Sheng-Yu Tseng *** and **Jun-Hao Fan**

Department of Electrical Engineering, Chang Gung University, Taoyuan City 33302, Taiwan; m0921014@cgu.edu.tw
* Correspondence: sytseng@mail.cgu.edu.tw; Tel.: +886-3-2118-800 (ext. 5706)

**Abstract:** Power quality is a critical issue in power systems. This paper proposes a bridgeless boost converter to increase the power factor of power systems using a utility line source for raising power quality. To reduce input and output current ripple, an interleaving manner is adopted in the proposed power system. When the interleaving bridgeless boost converter is used to implement power factor correction (PFC), it needs two bridgeless boost converters to process power during one switching cycle. In order to simplify the proposed bridgeless boost converter, two sets of switches in the conventional bridgeless boost one are integrated to reduce component counts. With this approach, the proposed bridgeless boost converter uses four switches to implement PFC features. Therefore, the proposed boost converter can increase conversion efficiency and decrease component counts, resulting in a higher conversion efficiency, lower cost and more simplicity for driving circuits. Finally, a prototype with a universal input voltage source (AC 90 V~265 V) under an output voltage of 400 V and a maximum output power of 1 kW has been implemented to verify the feasibility of the proposed bridgeless boost converter.

**Keywords:** bridgeless boost converter; power factor correction; PFC; interleaving manner

## 1. Introduction

The Internet of Things (IoT) is now widely applied to industrial, commercial and residential situations. Many sensors are adopted to construct the IoT. When a power supply generates a high-frequency noise, it will cause an abnormal signal of the sensor, leading to an error for IoT operations. Therefore, it needs a precision power supply and good power quality to supply a precision voltage level to control the system and avoid noise that affects its control functions. In addition, the power supply for IoT applications is required to have lighter, thinner, shorter and smaller features. As a result, a switching-mode converter is widely applied to these applications [1–11]. When a switching-mode converter adopts a utility line source as its power source, it will generate a seriously harmonic current pollution in the line source. In order to protect the line source from harmonic current pollution, a power factor corrector (PFC) is used in AC/DC power systems. It has to meet the various international power quality standards, such as International Electro-technical Commission (IEC) 61000-3-2 [12]. Thus, when an AC/DC converter uses PFC techniques to increase the power factor (PF), its input voltage can be made to be completely in phase with the input current one, implying an approximately unity power factor.

When an AC/DC power supply adopts active PFC to increase the PF, a boost converter is the most popular topology among the AC/DC converters. Since a boost converter is combined with a diode bridge rectifier to form the active PFC, losses from the diode bridge rectifier are significant, as shown in Figure 1. In particular, when the line source is at a lower input voltage and high output power, the conversion efficiency of the AC/DC converter is reduced. To improve the efficiency of the diode bridge rectifier, a bridgeless boost converter is adopted to reduce losses of diodes, as shown in Figure 2. Due to larger common-mode (CM) noise interference in the bridgeless boost converter shown in Figure 2, two diodes

are added into the bridgeless boost converter, as shown in Figure 3. When the AC/DC converter adopts the modified bridgeless boost converter, the conversion efficiency of the AC/DC one can be reduced. In order to increase conversion efficiency, diodes $D_{S1}$ and $D_{S2}$ are replaced by switches $M_{S1}$ and $M_{S2}$, as illustrated in Figure 4. As mentioned above, the AC/DC converters using the conventional bridgeless boost converter can reduce the power losses of diodes and increase conversion efficiency.

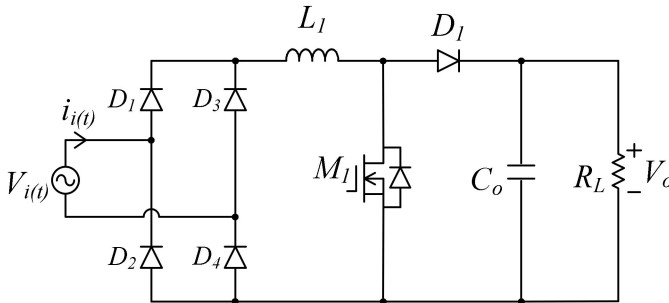

**Figure 1.** Schematic diagram of boost converter for power factor corrector (PFC) applications.

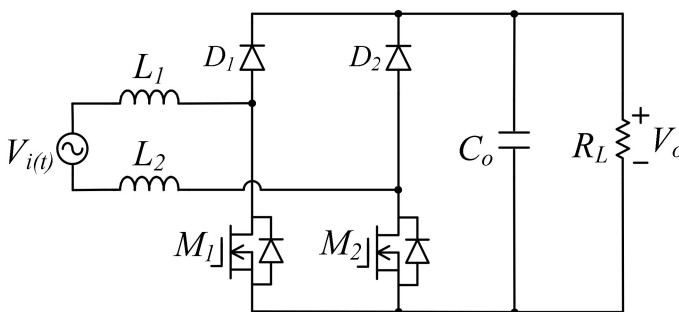

**Figure 2.** Schematic diagram of the bridgeless boost converter without low common-mode (CM) noise for PFC applications.

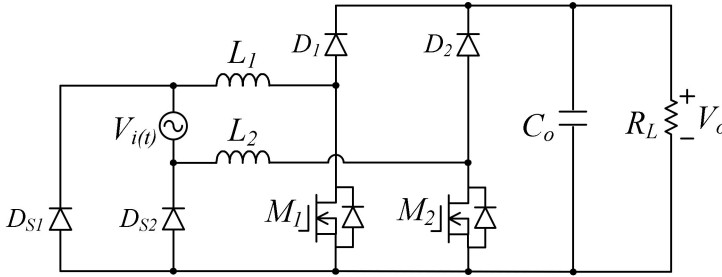

**Figure 3.** Schematic diagram of the bridgeless boost converter with low CM noise for PFC applications.

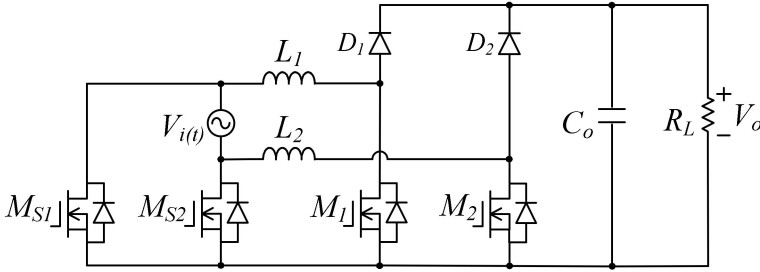

**Figure 4.** Schematic diagram of the modified bridgeless boost converter with low CM noise for PFC applications.

When a switching-mode converter is used in high-power applications, it will induce larger current ripples in input and output ports, resulting in a requirement for a larger passive filter. In order to reduce the passive component size, increase the output power level and decrease the current ripple, an interleaving circuit is usually adopted as an alternative solution in high-power applications. Many interleaving converters have been proposed, such as in [13–19]. When the AC/DC converter uses the conventional bridgeless boost converter for PFC applications, it can adopt its interleaving circuit to increase power processing capability, as shown in Figure 5. From Figure 5, it can be seen that the interleaving bridgeless boost converter can achieve a higher power factor, smaller current ripple and higher conversion efficiency. It is suitable for PFC applications.

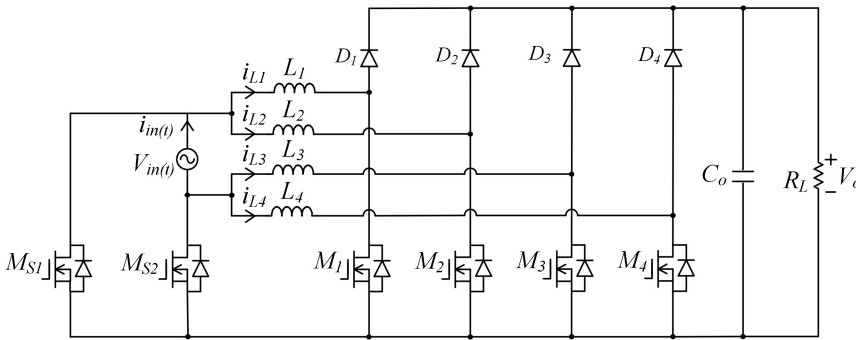

**Figure 5.** Schematic diagram of the modified interleaving bridgeless boost converter with a low CM noise for PFC applications.

When PFC adopts the interleaving bridgeless boost converter illustrated in Figure 5, it needs a more complex driving circuit. In order to further simplify the circuit, switches $M_1 \sim M_4$ shown in Figure 5 can be merged and replaced by switches $M_1$ and $M_2$, illustrated in Figure 6. According to performances of circuits shown in Figures 2–6, each PFC circuit can be used in different situations. Table 1 shows the performance comparisons among different PFCs with power flow through component paths. When power flows through each component, it will generate power loss in the component, resulting in lower conversion efficiency. The circuits, shown in Figures 2–4 are the single-phase topology. Their power level processing capability belongs in low- or medium-power applications. The circuits in Figures 5 and 6 are the two-phase topology. Their power level processing capacities are suitable for medium- or high-power applications. In Table 1, it can be seen that the converter in Figure 4 possesses the highest conversion efficiency, while the one in Figure 3 has lower conversion efficiency. When the converter topology adopts a two-phase manner, the proposed boost converter can reduce component counts, resulting in lower conversion efficiency compared with the modified bridgeless boost converter with low CM noise shown in Figure 5. Their conversion efficiency difference is about 1%. Therefore, the proposed boost converter possesses superiority in PFC applications. In Figure 6, the proposed interleaving bridgeless boost converter can implement PFC functions and reduce current ripple and increase conversion efficiency and power processing capability.

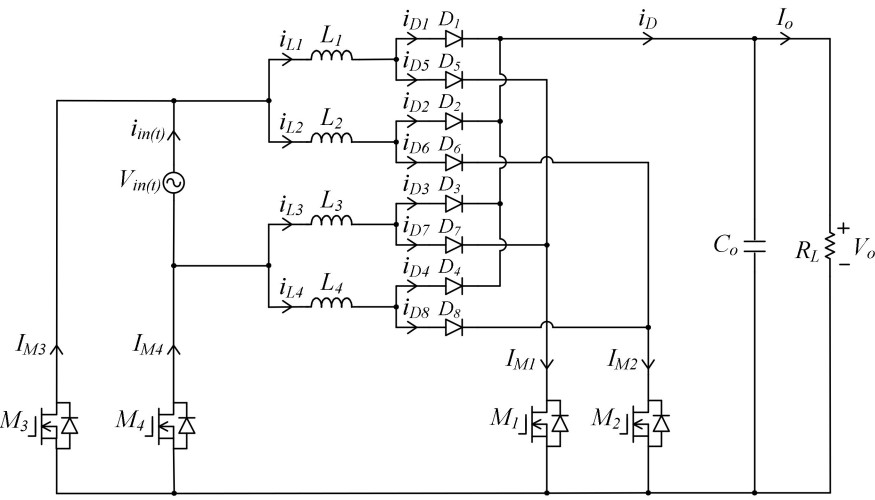

**Figure 6.** Schematic diagram of the proposed interleaving bridgeless boost converter.

**Table 1.** Performance comparison among different PFCs with power flow through component paths.

| Performances or Operational States | | Power Flow through Component Paths | | | | |
|---|---|---|---|---|---|---|
| | | Figure 2 Single Phase | Figure 3 Single Phase | Figure 4 Single Phase | Figure 5 Two Phase | Figure 6 Two Phase |
| The positive half period | inductor in the stored energy state | $M_1,M_2$ $L_1,L_2$ | $M_1,D_{S2}$ $L_1$ | $M_1,M_{S2}$ $L_1$ | $M_1,M_{S2}$ $L_1$ | $M_1,M_4$ $D_5,L_1$ |
| | | | | | $M_2,M_{S2}$ $L_2$ | $M_2,M_4$ $D_6,L_2$ |
| | inductor in the released energy state | $D_1,M_2$ $L_1,L_2$ | $D_1,D_{S2}$ $L_1$ | $D_1,M_{S2}$ $L_1$ | $D_1,M_{S2}$ $L_1$ | $D_1,M_4$ $L_1$ |
| | | | | | $D_2,M_{S2}$ $L_1$ | $D_2,M_4$ $L_2$ |
| The negative half period | inductor in the stored energy state | $M_1,M_2$ $L_1,L_2$ | $M_2,D_{S1}$ $L_2$ | $M_2,M_{S1}$ $L_2$ | $M_3,M_{S1}$ $L_2$ | $M_1,M_3$ $D_7,L_3$ |
| | | | | | $M_4,M_{S1}$ $L_2$ | $M_2,M_3$ $D_8,L_4$ |
| | inductor in the released energy state | $M_1,D_2$ $L_1,L_2$ | $D_2,D_{S1}$ $L_2$ | $D_2,M_{S1}$ $L_2$ | $D_3,M_{S1}$ $L_2$ | $D_3,M_3$ $L_3$ |
| | | | | | $D_4,M_{S1}$ $L_2$ | $D_4,M_3$ $L_4$ |
| Power level processing capability | | small | small | small | large | large |
| Input and output current ripple | | large | large | large | small | small |
| Efficiency | | higher | low | highest | highest | higher |

## 2. Derivation of the Proposed Converter

Figure 5 shows a schematic diagram of the conventional interleaving bridgeless boost converter for PFC applications. Since the utility line source is divided into the positive half period and the negative half one, its power flow is different during each half period operation. When the conventional bridgeless boost converter is operated in the positive half period, switch $Ms_2$ is turned on, and switches $M_1$ and $M_2$ are operated in an interleaving mode. If switches $M_1$ and $M_2$ are separately turned on, inductors $L_1$ and $L_2$, respectively, operate in the stored energy state. When switches $M_1$ and $M_2$ are separately operated in the turned-off state, inductors $L_1$ and $L_2$, respectively, operate in the released energy state through diodes $D_1$ and $D_2$. When the conventional bridgeless boost converter is operated

in the negative half period, switch $M_{s1}$ is turned on, and switches $M_3$ and $M_4$ operate in the interleaving mode. Inductors $L_3$ and $L_4$ can be worked in the stored and released energy states, respectively, through switches $M_3$ and $M_4$ or diodes $D_3$ and $D_4$. According to the operations mentioned above, the conventional bridgeless boost converter can achieve PFC function.

In order to simplify the circuit shown in Figure 5, switches $M_2$ and $M_4$ are replaced by one switch, $M_2$, and two diodes, $D_6$ and $D_8$, as shown in Figure 6. The switch $M_2$ plays a switching role and is the same as that of switches $M_2$ and $M_4$ shown in Figure 5. Diodes $D_6$ and $D_8$ are used to avoid a reverse current from inductors $L_2$ and $L_4$, respectively. Moreover, switches $M_1$ and $M_3$ can be replaced by switch $M_1$ and diodes $D_5$ and $D_7$, as shown in Figure 6. In Figure 6, it can be seen that switch $M_1$ can be, respectively, operated in the positive half and the negative half periods to replace switches $M_1$ and $M_3$, shown in Figure 5, when inductors $L_1$ and $L_3$ are worked in the stored energy state. Diodes $D_5$ and $D_7$ are adopted to avoid the reverse currents of inductors $L_1$ and $L_3$, respectively. In addition, diodes $D_1$ and $D_3$ are, respectively, operated in the released energy states of inductors $L_1$ and $L_3$ when switch $M_1$ is turned off.

## 3. Operational Principle of the Proposed Boost Converter

The proposed interleaving bridgeless boost converter is operated in a PFC manner. Its conceptual waveform is plotted in Figure 7 during a complete line period. When the proposed converter is operated during a complete line period, it can be divided into two operational periods: the positive and negative half periods. In the positive half period, switch $M_4$ is turned on and switch $M_3$ is turned off. In addition, switches $M_1$ and $M_2$ are operated in the interleaving manner. That is, their operation is out of phase by 180° for each switch. When the proposed converter operates in the negative half period, switch $M_3$ is turned on and $M_4$ is turned off. In the operational interval, switches $M_1$ and $M_2$ also operate in the interleaving manner. Since the operational principles of the proposed converter in the positive half period are the same as those in the negative half period, except that switch $M_4$ turned on in the positive half periods changed to $M_3$ turned on in the negative one, its operational principles are only described in this paper for the positive half period situation.

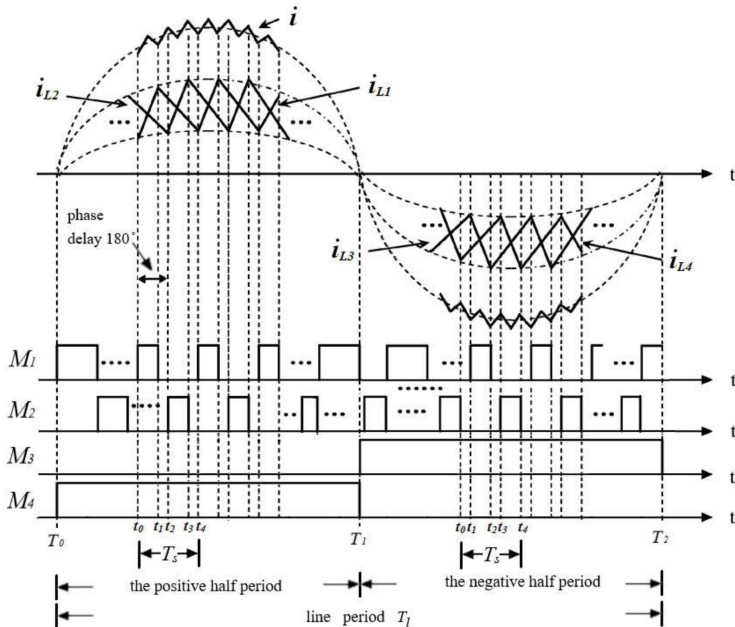

**Figure 7.** Conceptual waveforms of the proposed interleaving bridgeless boost converter during a complete line period.

When the proposed interleaving boost converter is operated in the positive half period, input voltage varies from 0 V to a maximum value and then from the maximum value to 0 V with a sine wave variation. The duty ratio of the switch in the proposed converter slowly decreases its value, which depends on the level of increase in the input voltage. When the input voltage is high enough, the duty ratio is less than 0.5 and inductor currents $i_{L1}$ and $i_{L2}$ operate in continuous conduction mode (CCM). Its conceptual waveform is shown in Figure 7. According to the operational principle of the proposed converter, its operational mode is divided into four modes. The equivalent circuit of each operational mode is illustrated in Figure 8. Each operational mode of the proposed converter is briefly described in the following section.

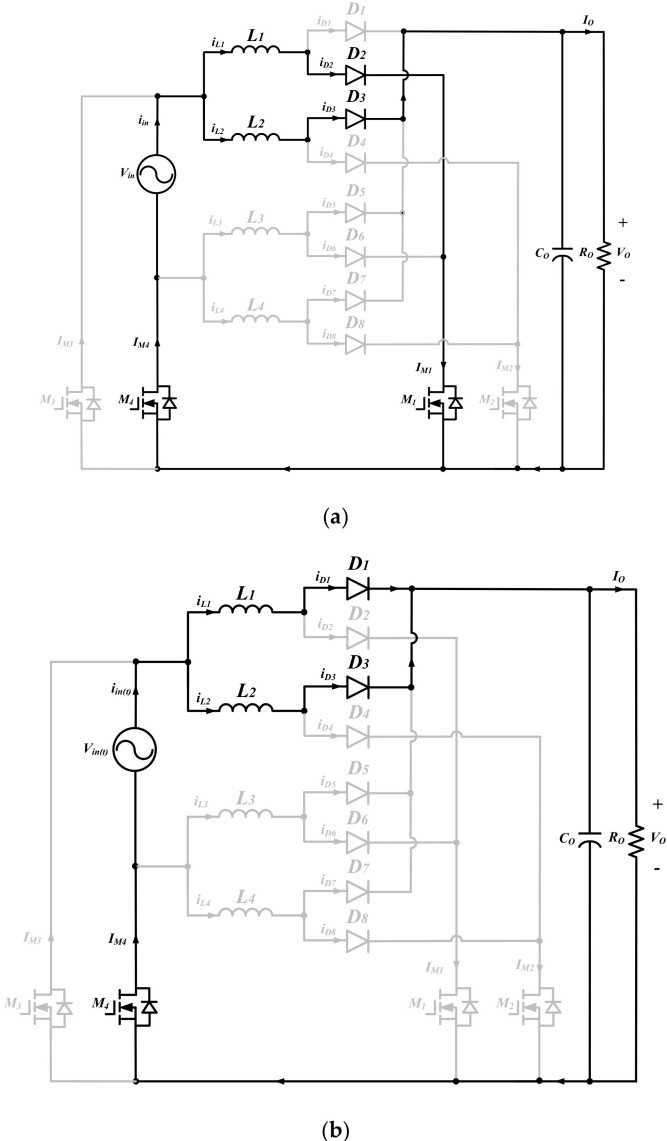

**Figure 8.** *Cont.*

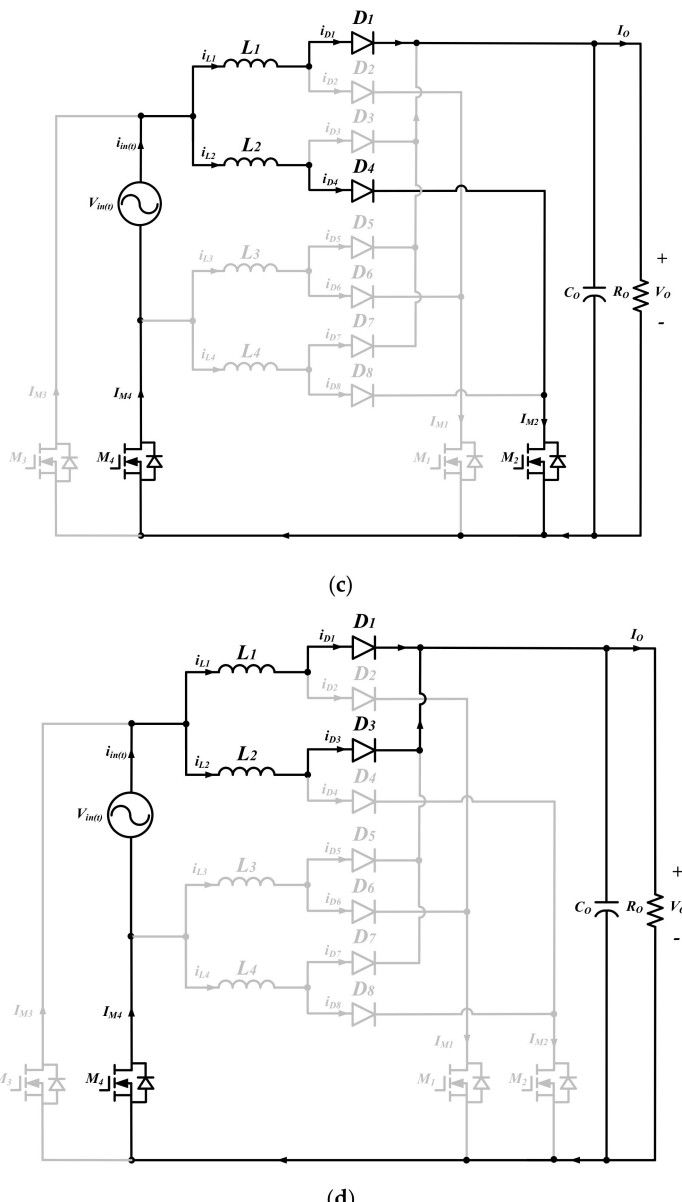

**Figure 8.** Equivalent circuit of the proposed interleaving bridgeless boost converter operated in the positive half period over one switching cycle. (**a**) Mode 1 ($t_0 \leq t \leq t_1$), (**b**) Mode 2 ($t_1 \leq t < t_2$), (**c**) Mode 3 ($t_2 \leq t < t_3$), (**d**) Mode 4 ($t_3 \leq t < t_4$).

Mode 1 (Figure 8a: $t_0 \leq t < t_1$): Before $t_0$, inductors $L_1$ and $L_2$ simultaneously work in the released energy states. Diodes $D_1$ and $D_2$ are forwardly biased. Currents $i_{L1}$ and $i_{L2}$ linearly decrease through $D_1$, load $R_0$ and switch $M_4$, and $D_3$, load $R_0$ and switch $M_4$, respectively. When $t = t_0$, switch $M_1$ is turned on and $M_4$ is kept in a turned-on condition. Diode $D_1$ is reversely biased and $D_2$ is forwardly biased, and inductor $L_1$ enters the stored energy state. In addition, inductor $L_2$ works in the released energy state through diode $D_3$. During this interval, inductor current $i_{L1}$ linearly increases and current $i_{L2}$ linearly decreases.

Mode 2 (Figure 8b: $t_1 \leq t < t_2$): At $t_1$, switch $M_1$ is turned off and $M_2$ is still in a turned-off state. Inductors $L_1$ and $L_3$ are in the released energy state through diodes $D_1$, $D_3$ and switch $M_4$, simultaneously. In this mode, currents $i_{L1}$ and $i_{L2}$ linearly decrease.

Mode 3 (Figure 8c: $t_2 \leq t < t_3$): When $t = t_2$, switch $M_2$ is turned on and $M_4$ is kept in the turned-on state. Inductor $L_2$ operates in the stored energy state through diode $D_4$ and switch $M_4$. Therefore, inductor current $i_{L1}$ linearly decreases and $i_{L2}$ linearly increases.

Mode 4 (Figure 8d: $t_3 \leq t < t_4$): At $t = t_3$, switch $M_2$ is turned off. Inductor $L_2$ changes the operation state from the stored energy state to the released energy state. Diodes $D_1$ and $D_3$ help inductors $L_1$ and $L_2$ turn to the released energy state, respectively. During this interval, currents $i_{L1}$ and $i_{L2}$ linearly decrease. When $t = t_4$, switch $M_1$ is turned on again. The new switching cycle starts.

## 4. Design of the Proposed Interleaving Bridgeless Boost Converter

The proposed interleaving bridgeless boost converter adopts the interleaving circuit to reduce input and output ripple currents. Its inductor ripple cancellation $k(D)$ is derived in this paper. For the design of the proposed converter, the determination of duty ratio $D$, inductors $L_1 \sim L_4$ and output capacitor $C_0$ is important. In addition, the selection of components $D_1 \sim D_8$, $M_1 \sim M_4$ is also described in this paper. Their design is briefly derived as follows.

(a)  Ripple current cancellation $K(D)$

The proposed interleaving boost converter can reduce the ripple currents of inductors $L_1 \sim L_4$. The ratio $K(D)$ of input ripple current $\Delta i_{in}$ to individual inductor ripple current $\Delta i_{L1}$ in the interleaving boost converter is plotted in Figure 9. When duty ratio $D$ is equal to or less than 0.5, $K(D)$ can determined by

$$K(D) = \frac{\Delta i_{\text{in}}}{\Delta i_{L1}} = \frac{1 - 2D}{1 - D} \tag{1}$$

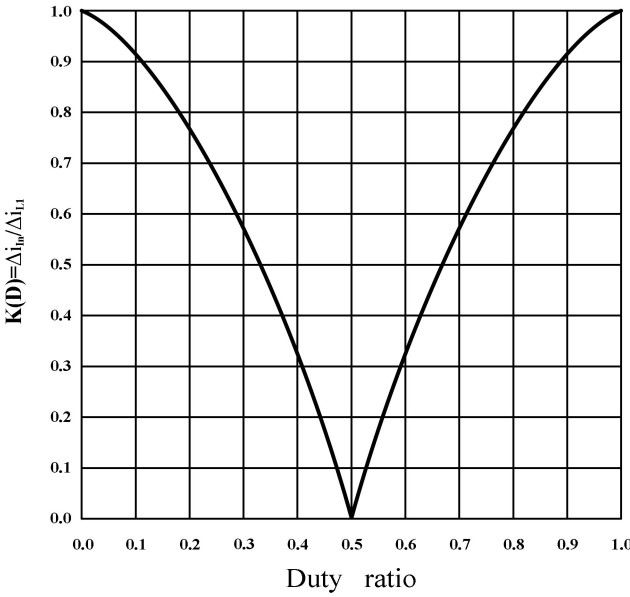

**Figure 9.** Plot of input inductor ripple current cancellation curve.

If $D$ is greater than 0.5, $K(D)$ can be derived by

$$K(D) = \frac{2D - 1}{D} \tag{2}$$

According to (1) and (2), when $D = 0.5$, $K(D) = 0$, input ripple current $\Delta i_{in}$ is equal to 0 A.

(b)  Duty ratio $D$

The input voltage range of the proposed boost converter is from AC 90 V to AC 265 V. When the input voltage is at low line, the duty ratio can be obtained by

$$D_{PLL} = \frac{V_o - V_{in\text{-}min} \times \sqrt{2}}{V_o} \tag{3}$$

where $V_{in\text{-}min}$ represents AC 90 V. If the input voltage is at high line, duty ratio $D_{PHL}$ can determined as

$$D_{PHL} = \frac{V_o - V_{in\text{-}max} \times \sqrt{2}}{V_o}, \tag{4}$$

where $V_{in\text{-}max}$ is equal to AC 265 V. Duty ratio $D$ can be varied from $D_{PHL}$ to $D_{PLL}$.

(c)　Inductors $L_1 \sim L_4$

Since the maximum inductor current $i_{L(max)}$ is at low line of the input voltage, inductors $L_1 \sim L_4$ can be determined for control within a desired range. In order to determine the values of inductors $L_1 \sim L_4$, the ripple current $\Delta i_L$ must be specified. When the proposed boost converter is operated at low line, $\Delta i_{L(max)}$ can be obtained by

$$\Delta i_{L(max)} = \frac{P_o \times \sqrt{2} \times 0.3}{V_{in\text{-}min} \times \eta \times K(D_{PLL})}, \tag{5}$$

where 0.3 means that the maximum input ripple current was set to 30% of the peak input current at low line, $\eta$ represents the conversion efficiency under a full-load condition and $K(D_{PLL})$ is the ratio of input current to inductor ripple current at the peak of low line operation. When $\Delta i_{L(max)}$ is determined, inductor $L_1 (= L_2 = L_3 = L_4)$ can be obtained as

$$L_1 = \frac{V_{in\text{-}min} \times \sqrt{2} \times D_{PLL} \times T_s}{\Delta i_{L(max)}}, \tag{6}$$

where $T_s$ is the switching period.

(d)　Output capacitor $C_o$

When the proposed boost converter is applied to PFC applications, output capacitor $C_o$ must sustain the output voltage $V_o$ at a desired value during loss of the line source under one line cycle. In general, when the line source faulty is during a line cycle, output voltage $V_o$ can be kept at and be greater than $0.75V_o$. According to the above requirement, output capacitor $C_o$ can be derived as

$$C_o \geq \frac{2P_o}{[V_o^2 - (0.75V_o)^2] \cdot f_l}, \tag{7}$$

where $f_l$ is the line frequency of the line source, and when output capacitor $C_o$ is determined, output ripple voltage $\Delta V_o$ can be expressed by

$$\Delta V_o = \frac{2P_o}{2\omega_l V_o C_o}, \tag{8}$$

where $\omega_l$ is equal to $2\pi f_l$.

(e)　Selection of switches and diodes

Figure 6 shows the schematic diagram of the proposed interleaving bridgeless boost converter. In order to determine the voltage and current ratings of components, the input voltage is in different situations. When the input voltage is in a high line situation, the voltage ratings of components in the proposed converter can be determined. The maximum voltage stresses of switches $M_1$ and $M_2$ can be determined by

$$V_{M1} = V_{M2} = V_o. \tag{9}$$

In addition, that of diodes $D_1 \sim D_8$ can be expressed by

$$V_{D1} = V_{D2} = V_{D3} = V_{D4} = V_{D5} = V_{D6} = V_{D7} = V_{D8} = V_o. \tag{10}$$

In a high line situation, the voltage stresses of switches $M_3$ and $M_4$ can be obtained: as

$$V_{M3} = V_{M4} = \sqrt{2}V_{in(max)}, \tag{11}$$

where $V_{in(max)}$ represents the input voltage level in a high line situation. When the input voltage is in a low line condition, rms currents $I_{M1(rms)}$ ($=I_{M2(rms)}$) of switch $M_1$ can be derived as

$$I_{M1(rms)} = I_{M2(rms)} = \left(\frac{P_o}{2V_{in\text{-}min}\eta}\right)\sqrt{1 - \frac{8\sqrt{2}V_{in\text{-}min}}{3\pi V_o}}, \tag{12}$$

where $P_o$ is the maximum output power, $V_{in\text{-}min}$ represents the rms value of input line voltage in a low line condition, $\eta$ is the conversion efficiency of the proposed converter and $V_o$ is the output voltage. Switch rms currents $I_{M3(rms)}$ ($= I_{M4(rms)}$) can be expressed by

$$I_{M3(rms)} = \left(\frac{P_o}{2V_{in\text{-}min}\eta}\right)\sqrt{\frac{1}{2} - \frac{4\sqrt{2}V_{in\text{-}min}}{3\pi V_o}}. \tag{13}$$

Additionally, the diode rms current $i_{D1(rms)}$ ($= i_{D2(rms)} = i_{D3(rms)} = i_{D4(rms)}$) is derived as

$$i_{D1(rms)} = \left(\frac{P_o}{2V_{in\text{-}min}\eta}\right)\sqrt{\frac{4\sqrt{2}V_{in\text{-}min}}{3\pi V_o}}. \tag{14}$$

The diode rms current $i_{D5(rms)}$ ($= i_{D6(rms)} = i_{D7(rms)} = i_{D8(rms)}$) is derived by

$$i_{D5(rms)} = \left(\frac{P_o}{2V_{in\text{-}min}\eta}\right)\sqrt{\frac{1}{2} - \frac{4\sqrt{2}V_{in\text{-}min}}{3\pi V_o}}. \tag{15}$$

Furthermore, the output capacitor rms current $I_{Co(rms)}$ is shown by

$$I_{Co(rms)} = \frac{P_o}{V_o\eta}\sqrt{\left(\frac{4\sqrt{2}V_o}{3\pi V_{in\text{-}min}}\right) - \eta^2}. \tag{16}$$

## 5. Experimental Results

The proposed interleaving bridgeless boost converter is shown in Figure 6. In order to verify the performance of the proposed converter, a prototype with the following specifications was implemented.

- Input voltage $V_{in}$: AC 90 V~265 V,
- Switching frequency $f_s$: 65 kHz,
- Output voltage $V_o$: 400 V,
- Maximum output current $I_{o(max)}$: 2.5 A and
- Maximum output power $P_{o(max)}$: 1 kW.

According to the equations mentioned above, the design values of key components of the proposed converter are listed in Table 2. The practical components are shown in the following section.

- Switches $M_1 \sim M_4$: IRFP460,
- Diodes $D_1 \sim D_8$: C3D10060,
- Inductances $L_1 \sim L_4$: 210 μH and
- Capacitor $C_o$: 1880 μF/450 V.

**Table 2.** Parameters of components in the proposed boost converter.

| Symbol | Calculation Value | | Practical Value | Relevant Parameters |
|---|---|---|---|---|
| | Equation | Value | | |
| $D_{PLL}$ | (3) | 0.70 | | $V_o = 400$ V; $V_{in\text{-}min} = 85$ V |
| $K(D_{PLL})$ | (2) | 0.57 | | $D = D_{PLL} = 0.7$ |
| $\Delta i_{L(max)}$ | (5) | 9.73 A | 6.16 A | $P_o = 1$ KW; $K(D_{PLL}) = 0.57$; $V_{in\text{-}min} = 85$ V; $\eta = 0.9$ |
| $L_1$ | (6) | 133 μH | 210 μH | $V_{in\text{-}min} = 85$ V; $D_{PLL} = 0.7$ $T_s = 15.38$ μs; $\Delta i_{L(max)} = 9.73$ A |
| $C_o$ | (7) | $\geq 476$ μF | 1880 μF | $P_o = 1$ KW; $V_o = 400$ V; $f_l = 60$ HZ |
| $\Delta V_o$ | (8) | 13.94 V | 3.53 V | $W_l = 2\pi f l = 377$ rad/s $P_o = 1$KW; $V_o = 400$ V |
| $V_{M1} = V_{M2}$ | (9) | 400 V | 500 V | $V_o = 400$ V |
| $V_{D1} = V_{D2} = \ldots = V_{D8}$ | (10) | 400 V | 600 V | $V_o = 400$ V |
| $V_{M3} = V_{M4}$ | (11) | 375 V | 500 V | $V_{in(max)} = 265$ V |
| $I_{M1(rms)} = I_{M2(rms)}$ | (12) | 5.64 A | 20 A | |
| $I_{M3(rms)} = I_{M4(rms)}$ | (13) | 6.54 A | 20 A | $M_1 \sim M_4$:IRFP460 (500 V/20 A) |
| $i_{D1(rms)}$ | (14) | 2.34 A | 10 A | $D_1 \sim D_8$:C3D10060 (600 V/10 A) $P_o = 1$ KW; $V_{in\text{-}min} = 85$ V |
| $i_{D5(rms)}$ | (15) | 3.99 A | 10 A | $H = 0.9$; $V_o = 400$ V |
| $I_{co(rms)}$ | (16) | 3.94 A | | |

The proposed interleaving bridgeless boost converter is proposed to achieve a higher PF and a lower input ripple current. Figure 10 shows the measured currents $i_{L1}$, $i_{L2}$ and $i_{in}$ waveforms of the proposed interleaving bridgeless converter under AC 90 V of input voltage. Figure 10a shows those waveforms with 10% of the full-load condition, while Figure 10b illustrates those waveforms with 100% of the full-load condition. In Figure 10, it can be seen that the proposed converter operates in discontinuous conduction mode (DCM) under 10% of the full-load condition, and operates in CCM under 100% of the full-load condition. Furthermore, the input ripple current can be reduced from a light load to a heavy load. Measured output voltage $V_o$ and current $I_o$ waveforms of step-load changes between 10% and 100% of the full-load condition with a duty ratio of 50% and a repetitive period of 1s are illustrated in Figure 11. Figure 11a depicts those waveforms under AC 220 V of the input voltage. From Figure 11, output voltage $V_o$ is regulated within a desired voltage range. Its value is limited within 1%.

The conversion efficiency of the proposed boost converter from a light load to a heavy load under the different input voltage is plotted in Figure 12. In Figure 12, when the input voltage is at a higher level, its conversion efficiency is higher than that of a lower input voltage from a light load to a heavy load. The maximum conversion efficiency is 96% under 80% of the full-load condition at AC 230 V of input voltage. From Figure 12, it can be seen that the conversion efficiency of the proposed boost converter is higher than 88% under different input voltages. Since the input voltage $V_{in}$ varies with the sine wave, it is difficult to evaluate the power loss of each component with an accurate method. In general, power loss analysis of PFC is usually adopted by simulation tools to obtain approximate power losses for the converter. Table 3 lists the parameters of selection components in the proposed boost converter. According to Equations (12)–(16), the rms currents of switches and diodes can be obtained from a light load to a heavy load at input voltage $V_{in}$ of AC 110 V. Table 4 shows the power loss of each semiconductor. In Table 4, power losses of switches include switching loss and conduction loss. Figure 13 shows the conceptual waveforms of switching loss during switch turn-on and turn-off transitions. In addition, the conduction loss of switches can be determined by $I_{M(rms)}^2 R_{ds(on)}$, where $I_{M(rms)}$ is the

average rms current of a switch and $R_{ds(on)}$ expresses the resistance of a switch in the conduction state. Power loss analysis of diodes is evaluated by $I_{D(rms)}V_F$, where $I_{D(rms)}$ is the average rms current of a diode and $V_F$ is its forward drop voltage. Switches $M_3$ and $M_4$ are turned on or turned off at the zero-crossing point. Therefore, their switching losses are equal to 0. Their power losses only consider conduction loss.

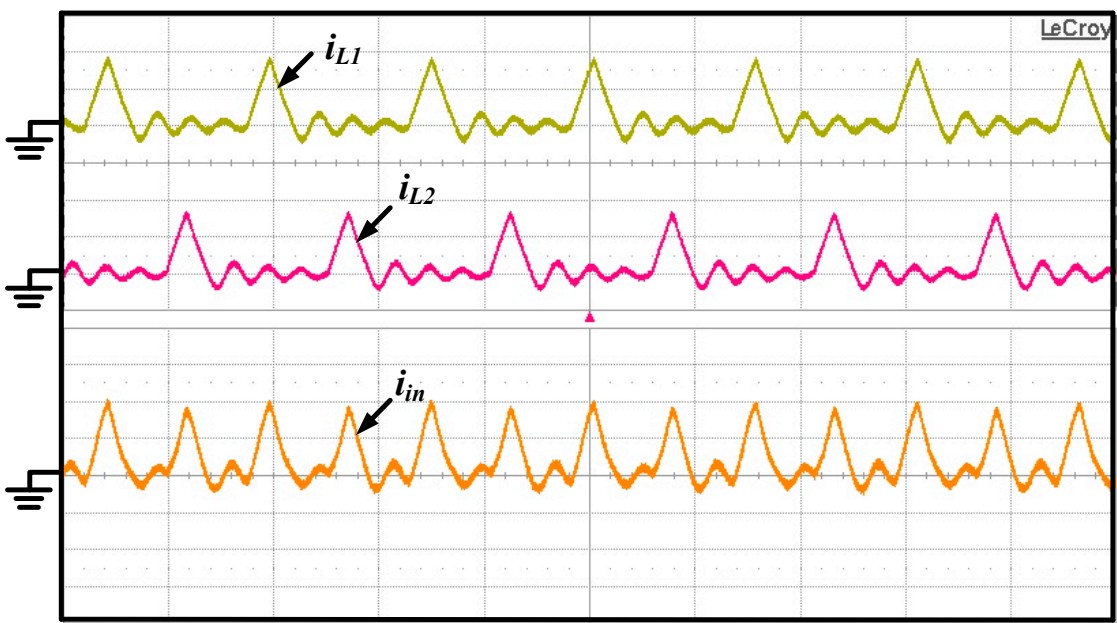

($I_{L1}$: 2 A/div, $I_{L2}$: 2 A/div, $i_{in}$: 2 A/div, time: 10 μs/div)

(**a**)

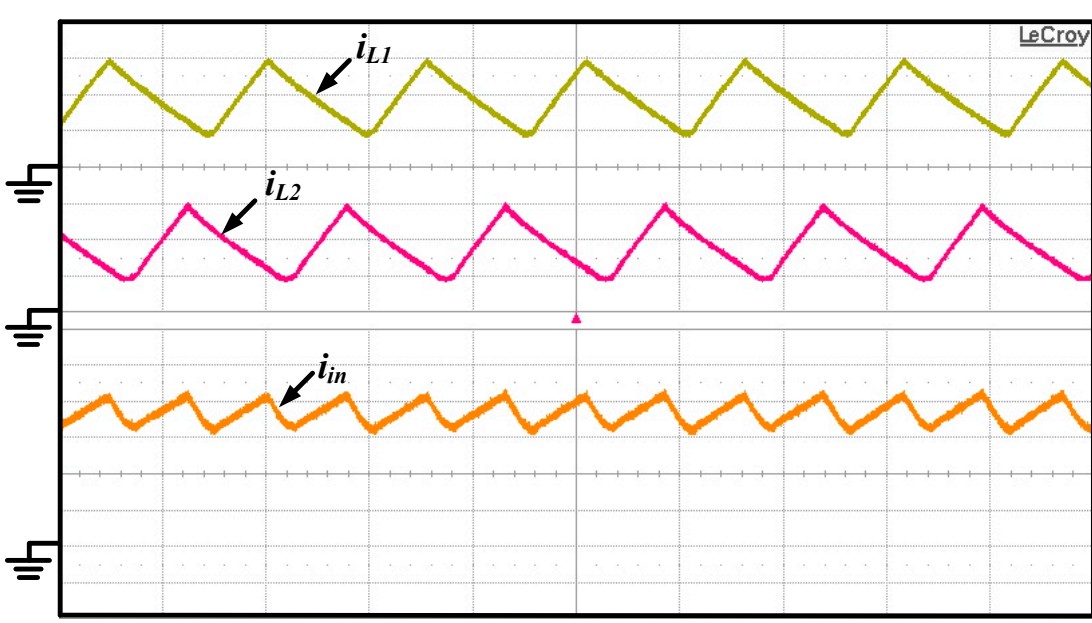

($I_{L1}$: 5 A/div, $I_{L2}$: 5 A/div, $i_{in}$: 5 A/div, time: 10 μs/div)

(**b**)

**Figure 10.** Measured currents $i_{L1}$ and $i_{L2}$ and $i_{in}$ waveforms of the proposed interleaving boost converter with (**a**) 10% of full-load condition, and (**b**) 100% of full-load condition under AC 90V of input voltage.

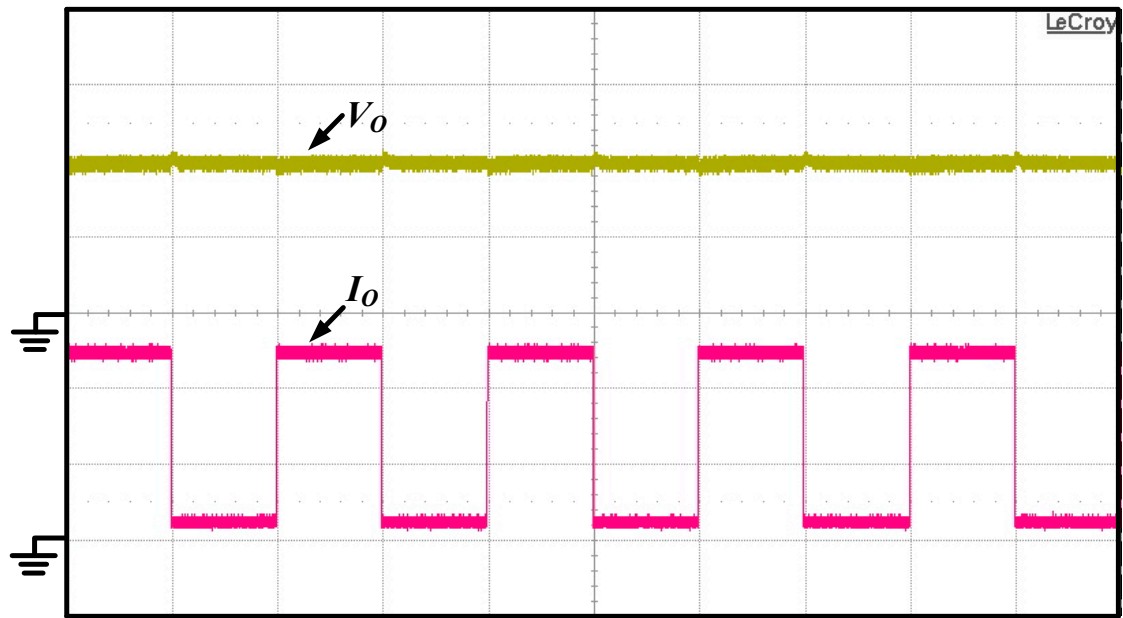

(*V*$_O$: 200 V/div, *I*$_O$: 1 A/div, time: 500 ms/div)

(**a**)

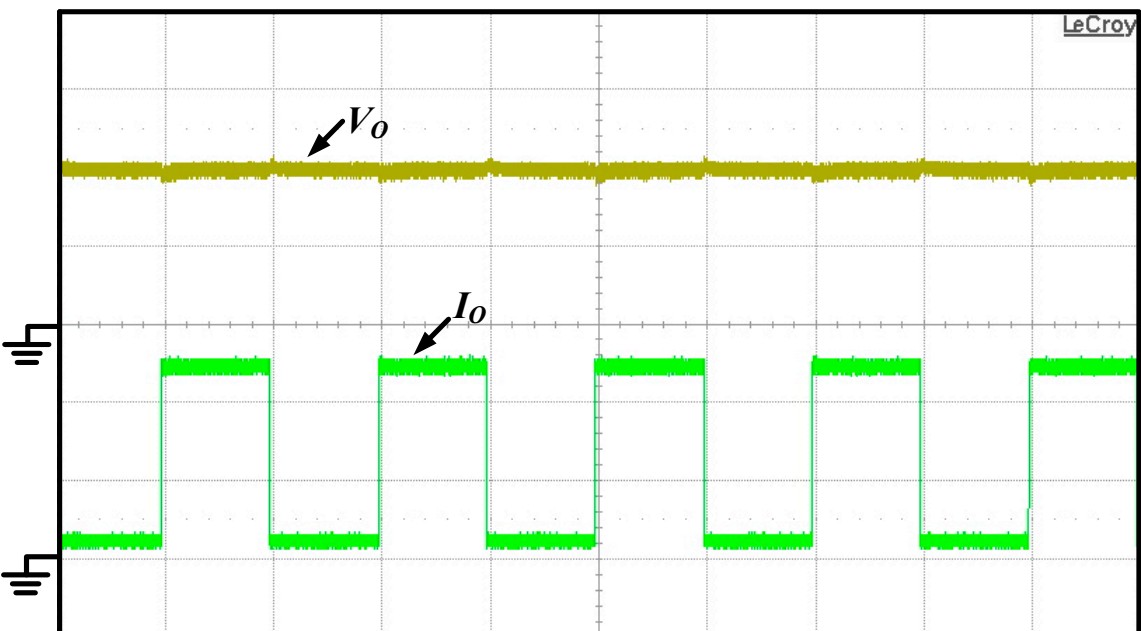

(*V*$_O$: 200 V/div, *I*$_O$: 1 A/div, time: 500 ms/div)

(**b**)

**Figure 11.** Measured output voltage $V_o$ and current $I_o$ waveforms of step-load changes between 10% and 100% of full-load conditions with duty ratio of 50% and repetitive period of 1 s: (**a**) under AC 110 V of input voltage, and (**b**) under AC 220 V of input voltage.

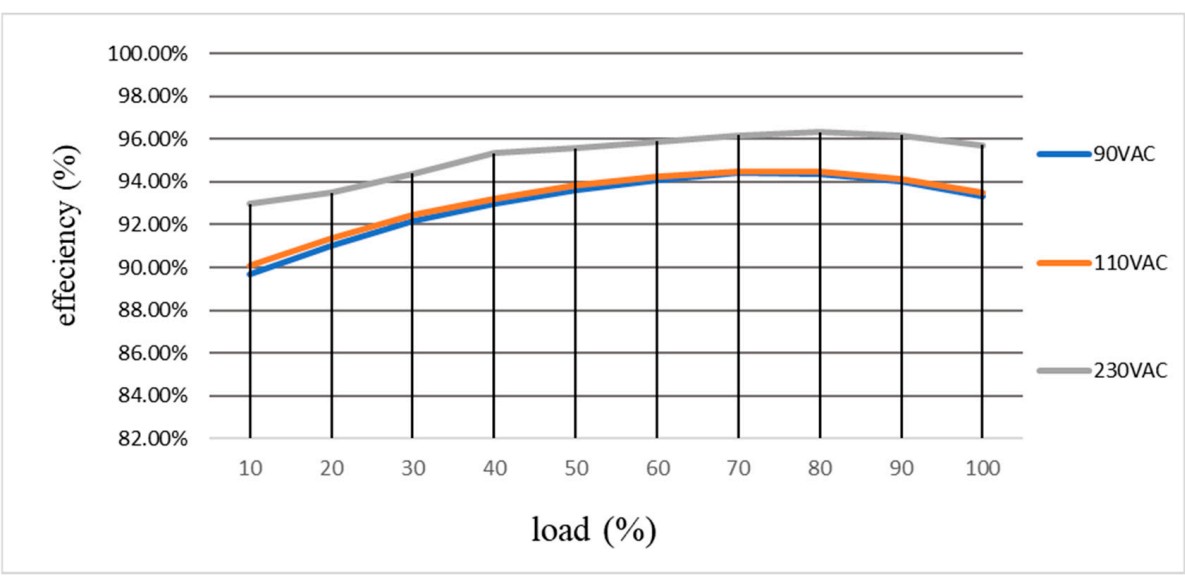

**Figure 12.** Plots of conversion efficiency of the proposed interleaving boost converter from light load to heavy load under the different input voltages.

**Table 3.** Key component parameters of the proposed converter.

| Component | Part Number | Voltage/Current Ratingsor Formula | Features | | |
|---|---|---|---|---|---|
| | | | Symbol | Parameter | Values |
| $M_1$, $M_2$ $M_3$, $M_4$ | IRFP640 | 500 V/20 A | $R_{DS(on)}$ | Drain–source turn-on resistance | 0.2~0.27 Ω |
| | | | $t_{on}$ | Turn-on transition time | 104 ns |
| | | | $t_{off}$ | Turn-off transition time | 150 ns |
| $D_1$~$D_8$ | C3D10060 | 600 V/10 A | $V_F$ | Forward voltage | 1.5 V |
| $L_1$~$L_4$ | Arnold MS-184060-2 | $B_m = \frac{u_o u_r \text{NI}_{pk}}{l_e}$ $u_o$: $4\pi*10^{-7}$H/m N: Turns $l_e$: Effective Magnetic Path Length(m) $I_{pk}$: Peak Current (A) | $u_r$ | Permeability | 60 |
| | | | $O_D$ | Outside diameter | 46.74 mm |
| | | | $I_D$ | Inside diameter | 24.13 mm |
| | | | $H_t$ | Height | 18.03 mm |
| | | | $A_e$ | Effective cross sectional area | 1.990 cm$^2$ |
| | | | $l_e$ | Effective magnetic path length | 10.743 cm |
| | | | $V_e$ | Effective core volume | 21.373 cm$^3$ |
| | | | $l_n$ | Approximate mean length of turn | 6.16 cm |
| | | | N | Turns | 32 |
| | | | AWG#18 | Wire gauge | Diameter 1.0 mm |
| | | | | $R_{dc}$: resistance | 21.4 mΩ/m |

**Table 4.** Semiconductor loss analysis for the proposed boost converter under input voltage of AC 100 V.

| Load (%) | Efficiency η (%) | Switches $M_1$ or $M_2$ Loss | | Switches $M_3$ or $M_4$ Loss | Diodes $D_1 \sim D_4$ | Diodes $D_5 \sim D_8$ |
| --- | --- | --- | --- | --- | --- | --- |
| | | Switching Loss $P_{stotal} = P_{sont}$ $P_{soff} = \frac{1}{2T_s} V_{DS}$ $(t_{on}I_{DB}+t_{off}I_{DP})$ (w) | Conduction Loss $P_{c1} = (I_{M1(rms)}{}^2 R_{ds(on)})$ (w) | Conduction Loss $P_{c2} = (I_{M5(rms)}{}^2 R_{ds(on)})$ (w) | Forward Drop Voltage Loss $P_{D1} = (i_{D1(rms)} V_F)$ (w) | Forward Drop Voltage Loss $P_{D5} = (i_{D5(rms)} V_F)$ (w) |
| 10 | 90 | 1.38 | 0.03 | 0.025 | 0.55 | 0.88 |
| 20 | 91 | 2.75 | 0.13 | 0.1 | 1.11 | 1.74 |
| 30 | 92.5 | 4.08 | 0.29 | 0.22 | 1.64 | 2.57 |
| 40 | 93 | 5.38 | 0.51 | 0.38 | 2.18 | 3.40 |
| 50 | 93.5 | 6.71 | 0.79 | 0.59 | 2.7 | 4.23 |
| 60 | 94 | 8.0 | 1.13 | 0.84 | 3.23 | 5.04 |
| 70 | 94.5 | 9.28 | 1.52 | 1.14 | 3.74 | 5.85 |
| 80 | 94.5 | 10.61 | 2.0 | 1.48 | 4.3 | 6.70 |
| 90 | 94 | 12.02 | 2.55 | 1.89 | 4.8 | 7.58 |
| 100 | 93 | 13.48 | 3.22 | 2.39 | 5.43 | 8.51 |

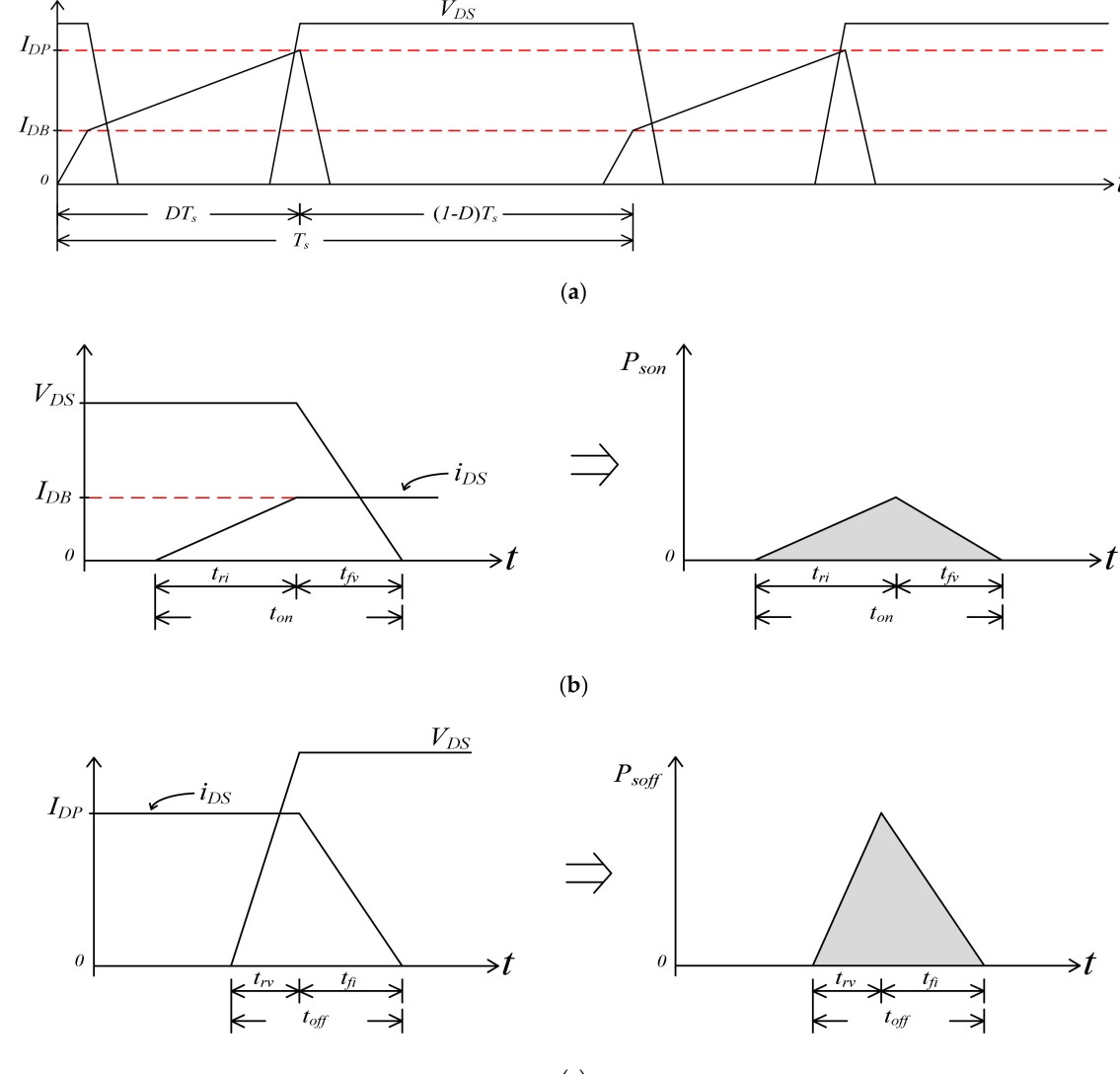

**Figure 13.** Conceptual waveforms of switching loss during (**a**) a complete switching cycle (**b**) switch turn-on transition, and (**c**) turn-off transition.

In the proposed boost converter, a major power loss is core loss $P_c$ and copper loss $P_{cp}$ in the inductors. Table 5 illustrates core loss $P_c$ and copper loss $P_{cp}$ of inductors $L_1{\sim}L_4$ of the proposed boost converter. Since inductors $L_1{\sim}L_4$ in the proposed boost converter are selected with super-MSS powder core manufactured by Arnold Magnetics LTD, its core loss curves are shown in Figure 14. The core loss must first obtain maximum flux density $B_m$, and then the core loss coefficient is determined, which is specified by Figure 14. Moreover, copper loss can be determined by $i^2_{L(rms)}R_{dc}$, in which $R_{dc}$ is the resistance of wire gauge. Table 6 illustrates the power loss analysis of the proposed boost converter under input voltage $V_{in}$ of AC110V. According to Table 6, it can be seen that when output power $P_o$ is less than 40% of the maximum output power $P_{o(max)}$, the calculation efficiency $\eta_c$ is higher than practical efficiency $\eta_p$. The reason for is that the stray losses do not include in the total power losses of the proposed converter, resulting in a lower practical efficiency. When output power Po is greater than 40% of the full-load condition, the proposed converter operates in CCM. Its practical peak currents of switches and inductors are less than those with the calculation method. Therefore, core losses of inductors and switching losses of switches with the calculation method are greater than the practical losses, meaning that the calculation efficiency $\eta_c$ is less than that of the practical efficiency $\eta_p$. Their difference is 2–3%. The calculation efficiency can be regarded as the reference efficiency.

**Table 5.** Parameters of core and core loss analysis for the proposed converter.

| Load (%) | Efficiency $\eta$ (%) | Input Current $i_{in}$ rms (A) | Core Loss and Parameters of Core | | | | | |
|---|---|---|---|---|---|---|---|---|
| | | | $i_{L1}(rms)$ rms (A) | $i_{L1}(pk)$ Peak Value (A) | Maximum Flux Density $B_m$ $B_m = \frac{u_o u_r N I_{pk}}{l_e}$ | Core Loss Coefficient (mw/cm$^3$) | Core Loss $P_C$ ($V_e$ = 21.373 cm$^3$) (w) | Copper Loss $P_{CP}$ ($l_e$ = 6.16 cm) (N = 32T) (w) |
| 10 | 90 | 1.01 | 0.35 | 0.71 | 171.2 G | 13 | 0.14 | ≒5.17 m |
| 20 | 91 | 2.0 | 0.7 | 1.41 | 340.02 G | 50 | 0.53 | ≒20.67 m |
| 30 | 92.5 | 2.95 | 1.03 | 2.09 | 504.01 G | 120 | 1.28 | ≒44.75 m |
| 40 | 93 | 3.91 | 1.37 | 2.76 | 665.58 G | 200 | 2.14 | ≒79.17 m |
| 50 | 93.5 | 4.86 | 1.70 | 3.44 | 829.56 G | 300 | 3.20 | 0.122 |
| 60 | 94 | 5.80 | 2.03 | 4.10 | 988.72 G | 420 | 4.49 | 0.173 |
| 70 | 94.5 | 6.73 | 2.36 | 4.76 | 1147.88 G | 580 | 6.2 | 0.235 |
| 80 | 94.5 | 7.7 | 2.70 | 5.44 | 1311.87 G | 710 | 7.59 | 0.307 |
| 90 | 94 | 8.7 | 3.05 | 6.15 | 1843.08 G | 860 | 9.19 | 0.392 |
| 100 | 93 | 9.78 | 3.42 | 6.91 | 1666.36 G | 980 | 10.47 | 0.493 |

**Table 6.** Power loss analysis for the proposed boost converter under input voltage of AC 110 V.

| Load (%) | Efficiency $\eta_p$ (%) | Switch Losses | | Diode Losses | | Total Core Losses $P_{TC}$ = ($P_C$ +$P_{CP}$) (w) | Total Power Losses $P_{loss}$ (w) | Calculation Efficiency $\eta_c$ (%) |
|---|---|---|---|---|---|---|---|---|
| | | $P_{TS1}$ = ($P_{M1}$ +$P_{M2}$) (w) | $P_{TS2}$ = ($P_{M3}$ +$P_{M4}$) (w) | $P_{TD1}$ = ($P_{D1}$ +$P_{D2}$+$P_{D3}$ +$P_{D4}$) (w) | $P_{TD2}$ = ($P_{D5}$ +$P_{D6}$+$P_{D7}$ +$P_{D8}$) (w) | | | |
| 10 | 90 | 2.82 | 0.05 | 2.2 | 3.52 | 0.56 | 9.15 | 9.16 |
| 20 | 91 | 5.76 | 0.2 | 2.22 | 3.48 | 2.12 | 13.78 | 93.6 |
| 30 | 92.5 | 8.74 | 0.44 | 3.28 | 5.14 | 5.12 | 22.72 | 93 |
| 40 | 93 | 11.76 | 0.76 | 4.36 | 6.8 | 8.56 | 32.24 | 92.5 |
| 50 | 93.5 | 15 | 1.18 | 5.4 | 8.46 | 13.28 | 43.32 | 92 |
| 60 | 94 | 18.26 | 1.68 | 6.46 | 10.08 | 18.64 | 55.12 | 91.6 |
| 70 | 94.5 | 21.6 | 2.28 | 7.58 | 11.7 | 25.76 | 68.92 | 91 |
| 80 | 94.5 | 25.22 | 2.96 | 8.6 | 13.4 | 31.6 | 81.78 | 90.7 |
| 90 | 94 | 29.14 | 3.78 | 9.6 | 15.16 | 38.32 | 96 | 90.4 |
| 100 | 93 | 33.4 | 4.78 | 10.86 | 17.02 | 43.84 | 109.9 | 90.1 |

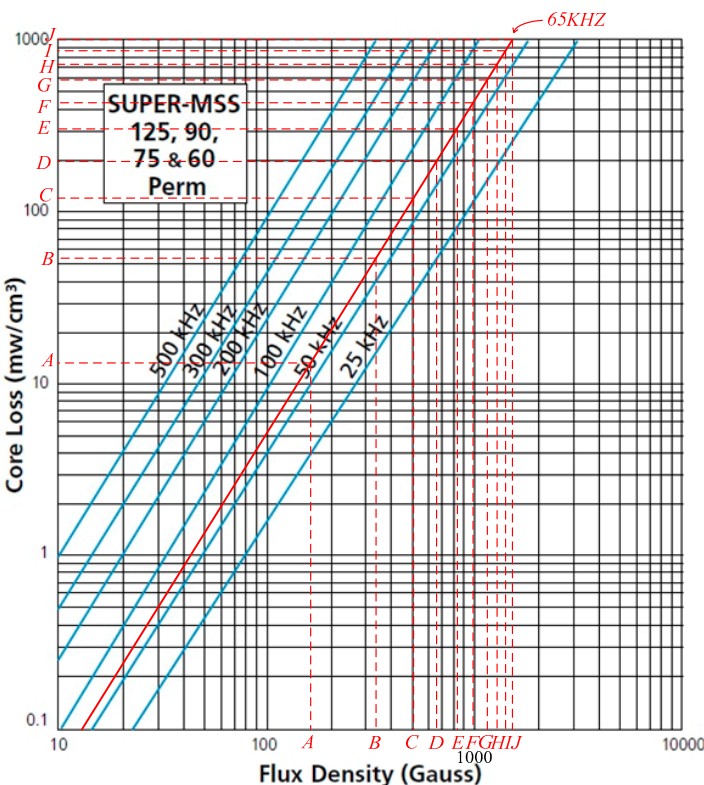

**Figure 14.** Core loss curves of inductors $L_1 \sim L_4$ manufactured by Arnold Magnetics LTD.

Figure 15 shows plots of the harmonic current of the proposed boost converter from a light load to a heavy load at different input voltages. From Figure 15, it can be seen that the harmonic current of the proposed converter from a light load to a heavy load under different input voltages can meet the requirements of IEC-6100-3-2 class A. In addition, plots of the power factor of the proposed converter from a light load to a heavy load under different input voltages are illustrated in Figure 16. With different input voltages, the power factor of the proposed converter from a light load to a heavy load is higher than 0.8. As mentioned above, the proposed interleaving bridgeless boost converter can implement a lower input ripple current, a higher conversion efficiency and a higher power factor. It is suitable for PFC applications.

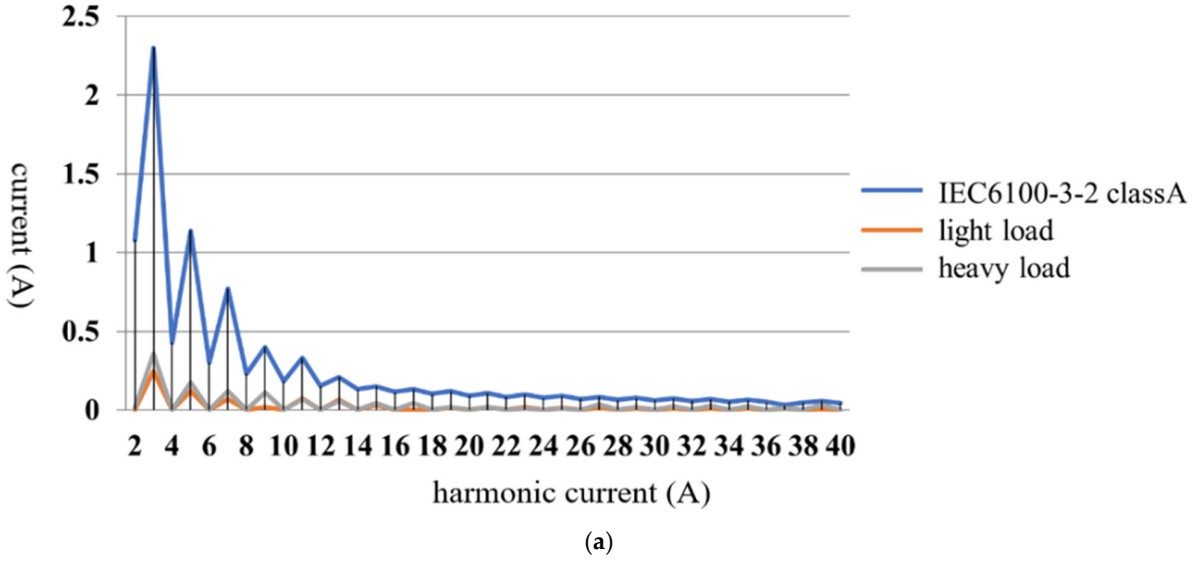

(**a**)

**Figure 15.** *Cont.*

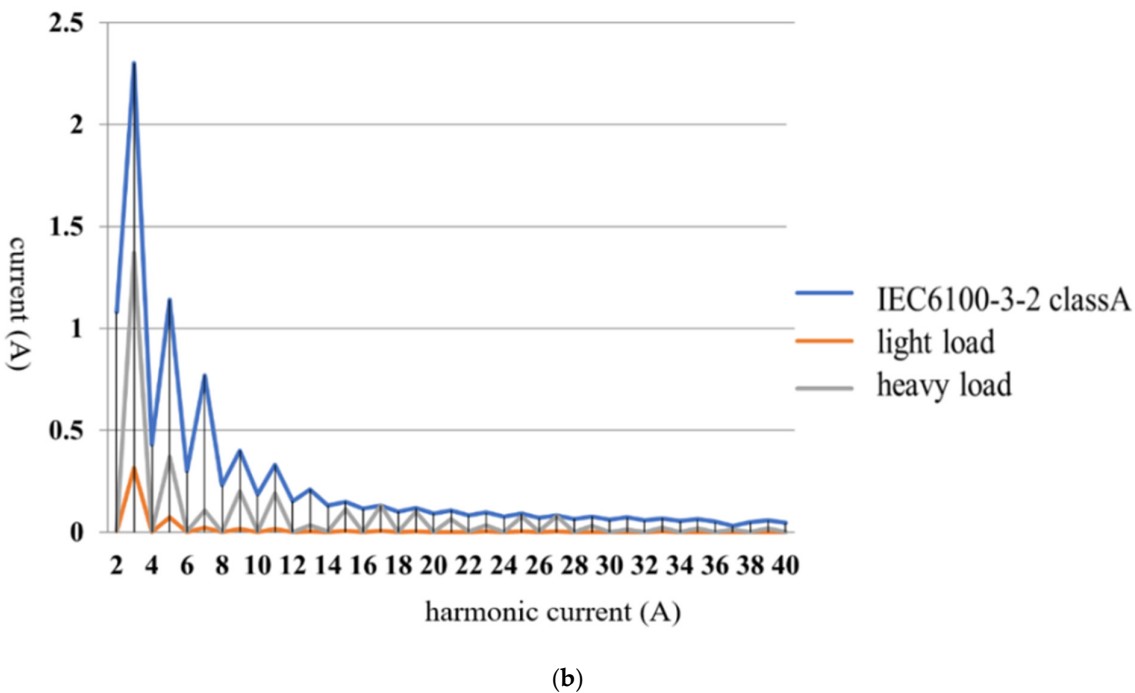

(**b**)

**Figure 15.** Plots of harmonic current of the proposed interleaving bridgeless boost converter from light load to heavy load (**a**) under input voltage of 110 V, and (**b**) under input voltage of 220 V.

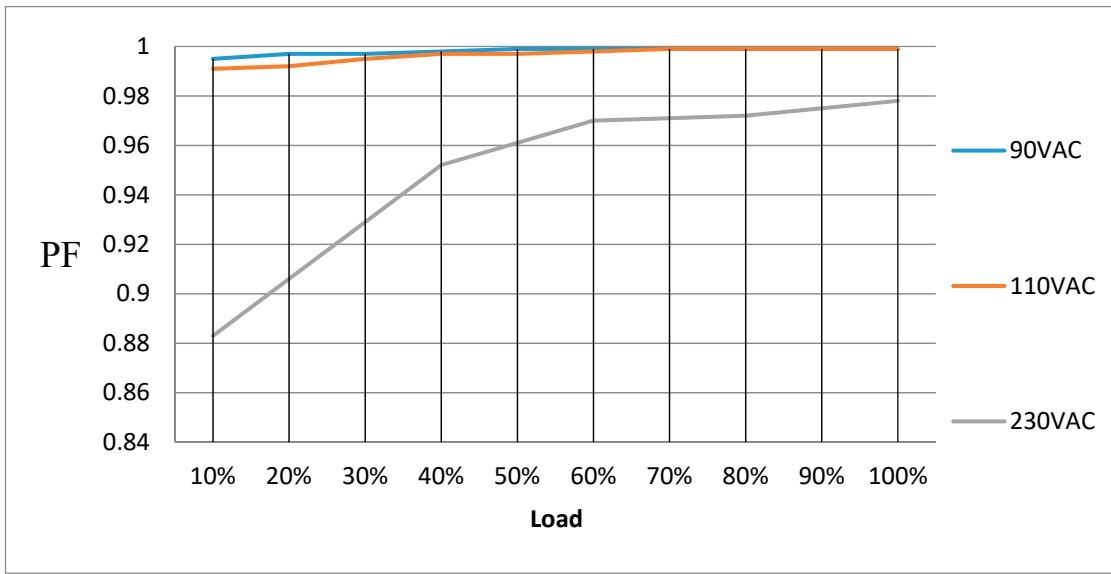

**Figure 16.** Plots of the power factor of the proposed interleaving bridgeless boost converter from light load to heavy load under different input voltages.

### 6. Conclusions

The proposed interleaving bridgeless boost converter is presented for PFC applications. The proposed converter adopts interleaving and bridgeless circuits to increase the power process capability and increase conversion efficiency, simultaneously. In this paper, the operational principle and design of the proposed converter have been described in detail. According to the experimental results of the proposed converter, input ripple current can be reduced and power factor can be increased. Furthermore, the proposed converter can also achieve a lower harmonic current and a higher conversion efficiency. Its harmonic current from a light load to a heavy load under different input voltages can meet the requirements

of IEC-6100-3-2 class A. The maximum conversion efficiency is about 96.4% under 80% of the full-load condition at AC 265 V of input voltage. In addition, the power factor from a light load to a heavy load under different input voltages is higher than 0.8. From the experimental results of the proposed interleaving bridgeless boost converter mentioned above, the proposed converter was implemented to verify its feasibility. It is suitable for PFC applications.

**Author Contributions:** Conceptualization, methodology, and writing—original draft preparation, S.-Y.T.; writing—review and editing, J.-H.F. All authors have read and agreed to the published version of the manuscript.

**Funding:** This research was funded by MOST in Taiwan, grant number MOST 109-2221-E-182-005.

**Conflicts of Interest:** The authors declare no conflict of interest.

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
