# Peer review of "Bridgeless Boost Converter with an Interleaving Manner for PFC Applications"

_electronics, doi:10.3390/electronics10030296_

Round 1

Reviewer 1 Report

This work presents a Bridgeless Boost Converter with the Interleaved Manner for PFC Applications. The authors have proposed and evaluated a topology, and it has been validated with experimental results. However, I have some comments about the work.
1) Due to the operation, from my understanding it seems that the semiconductors work at different voltage blocking conditions. I invite the authors to create a table with the rated operation values (voltage blocking) of each semiconductor.
2) Please include the losses of each semiconductor. The efficiency results shown in Fig.13 is useful to validate the complete converter behavior. However, the individual losses distribution gives more information about the operation on different operation points.
3) In fig. 13 at 10% of Load it seems that the efficiency goes to 0. Is it correct ? Could you please justify this behavior ?

Author Response

Author’s Reply to Review

Dear Reviewer NO. 1:

Taking this opportunity, the authors would like to express our sincere appreciation to your suggestion regarding our manuscript entitled “Bridgeless Boost Converter with the Interleaved Manner for PFC Applications”. The revised manuscript has completely mostly according to suggestions and comments of reviewer. It is modified by red word to highlight. In the following, the modified contents of the revised manuscript are briefly described.

  1. According to the comments from the reviwers, the revised manuscript adds 5 tables.
  2. According to the comments from the reviwers, the revised manuscript adds 2 figures.
  3. According to the comments from the reviwers, the revised manuscript deletes 2 figures, which Fig. 6(a) and Fig. 8 in the original manuscript.

Finally, we point out from yours and try our best to explain some vague discussion which did not convince you as follows:

  1. Due to the operation, from my understanding it seems that the semiconductors work at different voltage blocking conditions. I invite the authors to create a table with the rated operation values (voltage blocking) of each semiconductor.

<Author’s reply>

According to the operation principle of the proposed boost converter, voltage rating of each semiconductor is listed in Table 2. From Table 2, it can be seen that VM1(max)= VM2(max)= Vo, VD1(max)= VD2(max)= VD3(max) =VD4(max) =VD5max) =VD6(max) =VD7(max) =VD8(max) =Vo, and VM3(max)= VM4(max)= Vin(max). It is added in “5. Experimental results” of the revised manuscript.

  1. Please include the losses of each semiconductor. The efficiency results shown in Fig.13 is useful to validate the complete converter behavior. However, the individual losses distribution gives more information about the operation on different operation points.

<Author’s reply>

From Fig. 12, it can be seen that conversion efficiency of the proposed boost converter is higher than 88% under different input voltage. Since the input voltage Vin varies with sine wave, it is difficult to evaluate power loss of each component with an accurately method. In general, power loss analysis of PFC is usually adopted by simulation tools to obtain approximately power losses for power analysis of the one. Table 3 lists the parameters of selection component in the proposed boost converter. According to equations (12) ~ (16), the rms currents of switches and diodes can be obtained from light load to heavy load at input voltage Vin of AC110V. Table 4 plots the power loss of each semiconductor. In Table 4, power losses in switch include switching loss and conduction loss. Fig. 13 shows the conceptual waveforms of switching loss during switch turn-on and turn-off transitions. In addition, conduction loss of switch can be determined by , where IM(rms) is the average rms current of switch and Rds(on) expresses the resistance of switch in the conduction state. For power loss analysis of diode, it is evaluated by ID(rms)VF, where ID(rms) is the average rms current of diode and VF is its forward drop voltage. Switch M3 and M4 are turned on or turned off at the zero-crossing point. Therefore, their switching losses are equal to 0. Their power losses only consider conduction loss.

In the proposed boost converter, a major power loss is core loss Pc and copper loss Pcp in the inductors. Table 5 illustrates core loss Pc and copper loss Pcp of inductors L1 ~L4 of the proposed boost converter. Since inductors L1 ~L4 in the proposed boost converter are selected with super-MSS powder core manufactured by Arnold Magnetics LTD, its core loss curves is shown in Fig. 14. The core loss must first obtain maximum flux density Bm, and then determines core loss coefficient, which is specified by Fig. 14. Moreover, copper loss can be determined by , which Rdc is the resistance of wire gauge. Table 6 illustrates the power loss analysis of the proposed boost converter under input voltage Vin of AC 110V. According to Table 6, it can be seen that when output power Po is less than 40% of the maximum output power Po(max), the calculation efficiency ηc is higher than practical efficiency ηp. Its reason is that the stray losses does not include in the total power losses of the proposed one, resulting in a lower practical efficiency. When output power Po is greater than 40% of full-load condition, the proposed one is operated in CCM. Its practical peak currents of switches and inductors are less than those currents with calculation method. Therefore, core losses of inductors and switching losses of switches with calculation method are greater than the practical those losses, leading to the calculation efficiency ηc is less than that of the practical efficiency ηp. Their different value is with 2 ~ 3%. The calculation efficiency can be regarded as the reference efficiency. The above description is added in the “5. Experimental results” of the revised manuscript.

  1. In fig. 13 at 10% of Load it seems that the efficiency goes to 0. Is it correct ? Could you please justify this behavior ?

<Author’s reply>

According to Table 5 in the revised manuscript, the calculation efficiency under 10% of full-load condition is 91.6%. The stray loss of the proposed boost converter is about 1.6%. It can be explained that the practical efficiency is 90% under 10% of full load condition by calculation efficiency. It is listed in the “5. Experimental results” of the revised manuscript.

The form error has been modified in the revised manuscript.

Thanks again for your time and valuable comments.

With regards,

Sheng-Yu Tseng,

Associate Professor

Reviewer 2 Report

The manuscript presents an original solution of a bridgeless interlaced PFC converter. However, it seems that the manuscript was drafted too hurriedly and not very accurately. My main criticisms of the manuscript are as follows:

1) The introduction should further substantiate the advisability of the new solution in relation to the presented and other solutions of a PFC AC/DC boost converters. The results of the experiment should also be applied to other PFC AC/DC power converters.

2) I do not understand the need to refer in the introduction to the IoT (line 23). 

3) The size of figures 1 - 6 can be reduced 2 times without losing quality. The names of these drawings should also be corrected. In particular, please pay attention to Fig. 3 and Fig. 4 as well as Fig. 5 and Fig. 6a,b.

4) Is the conventional bridgeless converter shown in Figure 4? The reviewer believes that this is the modified converter presented in Fig.3, into which the diodes have been replaced by Ms switches to improve efficiency.

5) I do not see the need for two generally identical figures 6a and 6b (the markings can be standardised!) . In addition, Figure 6a incorrectly shows the input current.

6) Figures 7 and 9 are not fundamentally different. Only figure 7 should be left, with the switching cycle additionally marked. 

7) The size of Fig.9a-d may be reduced at least 2 times.

8)  In the opinion of the reviewer, the efficiency of the converter shown in Figure 5 will be higher than that of the tested converter (Fi.6). The paper should include a comparison of both solutions in terms of efficiency. The general statement about the need to use complexity driving circuits (lines 63-64) in the context of the availability of integrated driver technologies does not seem to be a proper argument.

9) Chapter 4 presents the formulae for designing the converter under test. The source from which these formulae are taken should be indicated, or the derivation of these formulae should be included, e.g. as an appendix.

10) Please indicate how the formulas (5) - (16) were used to design the prototype (chapter 5) - in particular, the compliance of formulae (6) and (7) with the experimental values for key components L and C 

Author Response

Dear Reviewer:

Taking this opportunity, the authors would like to express our sincere appreciation to your suggestion regarding our manuscript entitled “Bridgeless Boost Converter with the Interleaved Manner for PFC Applications”. The revised manuscript has completely mostly according to suggestions and comments of reviewer. It is modified by red word to highlight. In the following, the modified contents of the revised manuscript are briefly described.

  1. According to the comments from the reviwers, the revised manuscript adds 6 tables.
  2. According to the comments from the reviwers, the revised manuscript adds 2 figures.
  3. According to the comments from the reviwers, the revised manuscript deletes 2 figures, which Fig. 6(a) and Fig. 8 in the original manuscript.

Finally, we point out from yours and try our best to explain some vague discussion which did not convince you in the attached file.

The form error has been modified in the revised manuscript.

Thanks again for your time and valuable comments.

With regards,

Sheng-Yu Tseng,

Associate Professor

Reviewer 3 Report

This article presents bridgeless boost converter with a target to increase power factor of power system. 

The strength parts of this study is given by the design of the proposed interleaving bridgeless boost converter both with experimental results.

The slightly weaker part of the paper is given by the research of the state of the art of this topic.

Author Response

(The authors gave the same response as above.)

Round 2

Reviewer 1 Report

Thank you very much for the efforts improving the proposed work. I suggest to accept the paper in the present form.

Reviewer 2 Report

I have no additional comments